# Automated ranking of chest x-ray radiological finding severity in a binary label setting

**Matthew Macpherson**[1]      MATTHEW.MACPHERSON@WARWICK.AC.UK
**Keerthini Muthuswamy**[2]      KEERTHINI.MUTHUSWAMY@GSTT.NHS.UK
**Ashik Amlani**[2]      ASHIK.AMLANI@GSTT.NHS.UK
**Vicky Goh**[3,4]      VICKY.GOH@KCL.AC.UK
**Giovanni Montana**[1,5]      G.MONTANA@WARWICK.AC.UK

[1] *Warwick University, Coventry CV4 9AL*

[2] *Guy's and St Thomas' NHS Foundation Trust, London, UK*

[3] *School of Biomedical Engineering & Imaging Sciences, King's College London, London, UK*

[4] *Department of Radiology, Guy's and St Thomas' NHS Trust, London, UK*

[5] *The Alan Turing Institute, London, UK*

**Editors:** Accepted for publication at MIDL 2024

## Abstract

Machine learning has demonstrated the ability to match or exceed human performance in detecting a range of abnormalities in chest x-rays. However, current models largely operate within a binary classification paradigm using fixed decision thresholds, whereas many clinical findings can be more usefully described on a scale of severity which a skilled radiologist will incorporate into a more nuanced report. This limitation is due, in part, to the difficulty and expense of manually annotating fine-grained labels for training and test images versus the relative ease of automatically extracting binary labels from the associated free text reports using NLP algorithms. In this paper we examine the ability of models trained with only binary training data to give useful abnormality severity information from their raw outputs. We assess performance using manually ranked test sets for each of five findings: cardiomegaly, consolidation, paratracheal hilar changes, pleural effusion and subcutaneous emphysema. We find the raw model output predicts human-assessed severity ranking with Spearman's rank coefficients between 0.563 - 0.848. Using patient age as an additional variable with full ground truth ranking available, we compare a binary classifier output against a fully supervised RankNet model, quantifying the increase in training data required for equivalent performance. We show that model performance is improved using a semi-supervised approach supplementing a smaller set of fully supervised images with a larger set of binary labelled images.

**Keywords:** Chest x-ray, severity assessment, ranking, weakly supervised.

## 1. Introduction

Machine learning offers impressive capabilities in medical image classification, and has recently been shown to match or exceed human performance across a wide range of clinical findings in comprehensive multi-label settings (Wu et al., 2020; Seah et al., 2021; Dicente Cid et al., 2023). However, machine learning approaches to abnormality classification generally operate within a binary classification paradigm, with pre-determined output thresholds for the presence or absence of an abnormality. This omits valuable clinical information since

many abnormalities are usefully described on a spectrum of severity. A continuous measure of finding severity would be beneficial to both triage and disease progression monitoring and more closely replicate the skills of an experienced radiologist.

This limitation is largely due to the difficulty of obtaining sufficient quantities of granular training labels. Whereas binary labels can be obtained reliably from the original plain text clinical reports using natural language processing (NLP) algorithms (Cornegruta et al., 2016; Irvin et al., 2019; Dicente Cid et al., 2023), granular severity labels generally require labour intensive hand labelling by radiologists. Estimation of quantitative parameters using small sets of manually annotated training data has been explored for e.g. cardiothoracic ratio (CTR) as a measure of cardiomegaly (Li et al., 2019; Que et al., 2018; Chamveha et al., 2020) and Cobb angle (Chen et al., 2019; Tu et al., 2019; Kim et al., 2020) as a measure of scoliosis severity. Other work considers severity assessment using training data subjectively assigned to severity buckets by expert annotators, including for COVID19 (Zhu et al., 2020; Signoroni et al., 2021), tumour malignancy level (Mukherjee et al., 2020), edema progression (Horng et al., 2020), or general infection severity (Chandra et al., 2022). In the binary label setting, (Cohen et al., 2020) investigates predictive models of COVID19 severity based on fitting linear regression models to the outputs of a classifier trained on non-COVID19 datasets, demonstrating that the pre-trained outputs for lung opacity and related findings are predictive of COVID19 geographic extent and mortality.

The focus of this study is the possibility of estimating the severity of radiological findings without direct, granular ground truth training data. Instead, we have access only to binary labels indicating the presence or absence of findings, regardless of severity. Our objective is to explore whether models trained solely on these binary outcomes can implicitly capture information related to the severity of findings when they are present. To facilitate this, we engaged radiologists to manually evaluate the severity of historical chest x-rays through a method based on pairwise comparisons. This approach requires radiologists to compare two images at a time, simplifying the process of establishing a comprehensive ranking of severity across a dataset. Crucially, these rankings were utilized solely for the purpose of evaluating our hypothesis and were not incorporated into the training data. This study design allows us to empirically investigate the capability of binary-trained predictive models to infer nuances of severity, offering insights into their potential utility beyond simple binary classification tasks.

Our main contributions are:

- We provide a novel investigation of the continuous logits outputs of a pre-trained chest x-ray multi-label classifier, trained only with NLP-generated binary labels, as a predictor of severity ranking across five radiological findings. We evaluate performance using test sets ranked in order of presentation severity by two radiologists.

- We compare this approach with a fully supervised pairwise ranking model, RankNet (Burges et al., 2006), using patient age as a test variable with complete ground truth ranking available, to assess the relative efficacy of binary labelled versus fully supervised training data.

- We show that adding a small fraction of annotated data to a large volume of binary training data leads to stronger severity prediction performance in a combined model.

## 2. Methods

### 2.1. Data used

This work is based on a proprietary set of 1.9 million anonymised chest x-rays obtained under a research partnership with three NHS hospital networks (University Hospitals Coventry and Warwickshire NHS Trust, University Hospitals Birmingham NHS Foundation Trust, and University Hospitals Leicester NHS Trust) collectively representing six hospitals. The images comprise all adult ($> 15$ years) frontal images obtained from the PACS of these hospitals for patients from all clinical settings within the hospitals. Images were assigned binary labels for 37 radiological findings using a custom NLP model operating on the original free-text reports, with patient age obtained directly from the DICOM metadata. This image set was used to train and evaluate the age ranking models presented, with images were split into a training set of 1,694,921 images, an 89,238 image validation set and a test set of 103,328 images. In addition, a separate set of $1,427$ images was exhaustively manually annotated by a panel of three radiologists to act as a gold standard ground truth test set. Full details of the dataset acquisition, preprocessing and underlying patient populations can be found in (Dicente Cid et al., 2023).

### 2.2. Ranked test set development

Test sets of chest x-rays for each of four abnormalities were constructed, fully ranked in order from least to most severe presentation of the named abnormality. In each case, 50 images were sampled randomly from all images in the ground truth image set labelled positively with that finding (i.e. also potentially containing co-morbidities). Each set was then ranked in order of severity of the given finding independently by two radiologists. To facilitate the ranking process we used a pairwise approach using the merge sort algorithm to dynamically select pairs for comparison; such an approach offers a theoretical complexity of $\mathcal{O}(n \log n)$, minimising the number of individual comparisons required to rank each image set to approximately 200 pairwise comparisons. A web interface (see appendix Figure 4) was developed to present the image pairs to the radiologists at 1024x1024 resolution, who then selected the image with the higher perceived degree of severity. Following completion of the ranking algorithm, the radiologist was given the opportunity to review the ordered images to correct ranking errors. Ranking and reviewing each set of 50 images took approximately 1 hour for each radiologist. In cases where there was a difference in rankings of 10 or more between operators, that image was left out of the final test set. After eliminations, we retained 41 ranked images each for pleural effusion and consolidation, 50 for subcutaneous emphysema and 32 for paratracheal hilar changes. The retained images were scored with the mean rank position of the two radiologists to reach a final ranking measure.

Two test sets were also constructed with alternative ranking approaches. Firstly, cardiomegaly severity was assessed directly as the measured cardiothoracic ratio (CTR), the measured ratio of heart diameter to maximum thoracic width, giving a quantitative ranking for each image, yielding 173 ranked images. Secondly, patient age converted to a binary 'young/old' label through a boundary at 55 years was used as analogous to a radiological finding with full ground truth ranking available, allowing the entire dataset to be used as training/test data in either binary or ranked modes.

### 2.3. Approaches to modelling severity

#### 2.3.1. BINARY CLASSIFIER LOGIT OUTPUT

We first considered the logit output from a binary classifier as a predictor for the severity of the finding in question, on the assumption that higher probabilities of a finding being present might correspond to more obvious presentation/more severe symptoms. For the radiological findings we used an open source pre-trained classifier, X-Raydar (Dicente Cid et al., 2023), to generate the logit outputs. For patient age, we custom trained a new binary classification model with our training data using binary cross-entropy (BCE) loss. For this model we used an EfficientNet-B3 CNN backbone (Tan and Le, 2019), which has reported strong performance in the literature with a relatively light computational overhead.

#### 2.3.2. SEMI-SUPERVISED RANKING LOSS

We aimed to investigate if adding varying amounts of fully ranked training data to the full binary labelled training set could improve severity prediction performance, by combining the binary classification task with a fully supervised 'fine-tuning' ranking task. For this task we used patient age as a test variable; a binary label for patient age $> 55$ years was available for all training images, alongside randomly sampled fully age-ranked training sets of 1,000 to 100,000 images. This was intended to be analogous to binary labels for a radiological finding alongside a limited number of images hand-ranked by clinical experts.

In order to incorporate this ranked data into the training process, a pair-wise ranking loss $\mathcal{L}_{\text{RankNet}}$ was added to the BCE classification loss $\mathcal{L}_{\text{BCE}}$. We implemented RankNet (Burges et al., 2005) as the pairwise ranking algorithm. In this pairwise ranking approach, for each pair the difference between the image embeddings is passed through a sigmoid activation to produce a probability of correct ordering; this is compared with the true ordering via a BCE loss summation:

$$\mathcal{L}_{\text{RankNet}} = -\frac{1}{N} \sum_{i=1}^{N} \sum_{j=1}^{N} y_{ij} \log(p(y_{ij}) + (1 - y_{ij}) \log(1 - p(y_{ij})) \tag{1}$$

where:

$$p(y_{ij}) = \frac{1}{1 + exp(\mathbf{M}(x_i) - \mathbf{M}(x_j))} \tag{2}$$

where $x_i$ and $x_j$ are the two images and $\mathbf{M}$ the ranking model. The total semi-supervised loss is the sum of the binary classifier loss with this ranking loss:

$$\mathcal{L}_{\text{SEMI}} = \mathcal{L}_{\text{BCE}} + \lambda.\mathcal{L}_{\text{RankNet}} \tag{3}$$

where $\lambda$ is a hyperparameter to balance the loss components. The minibatches for the binary and ranking loss components were drawn independently from the binary labelled and fully labelled subsets respectively. Training parameters are detailed in appendix B.

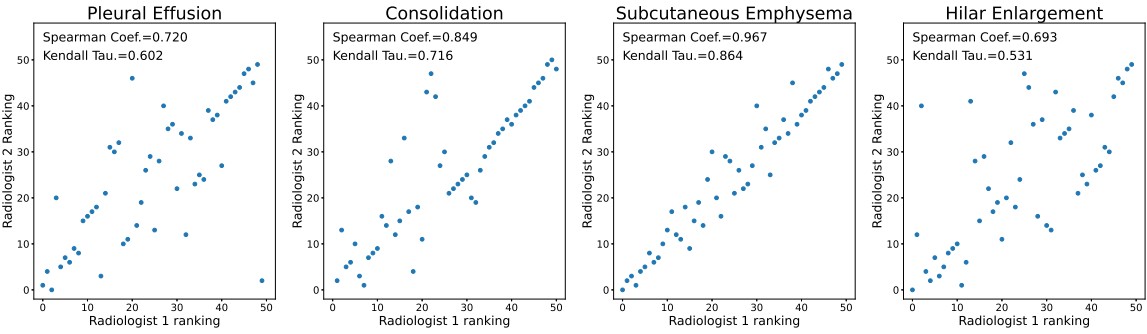

Figure 1: Inter-operator agreement scatter plots of test set severity rankings between the two radiologist annotators. Rankings show obvious correlation particularly at the extremes, albeit with noticeable major outliers. Subcutaneous emphysema and consolidation exhibit the strongest levels of agreement.

## 3. Results

### 3.1. Consistency of radiologist rankings

The reliability of the ranked test sets was assessed by comparing the independent rankings produced by the two radiologists. The relative rankings for each set are shown in Fig.1. There was a reasonable level of consistency between the operators, with the strongest relationship in subcutaneous emphysema (Spearman coefficient 0.967, Kendall Tau 0.864) and the weakest for hilar enlargement (0.693, 0.531), with obvious outliers which may result from disagreements early in the merge sort algorithm process leading to significantly different final orderings. In the appendix Fig.5 we present example images of increasing assessed severity for each of the findings.

### 3.2. Binary classifier logits outputs as severity predictors

In Fig.2 we show plots of the named X-Raydar logits output against the mean ranking of that finding in the relevant test set. Positive Spearman's rank coefficients from 0.563 (pleural effusion) to 0.848 (cardiothoracic ratio) are seen for the findings indicating that the classifier output has some predictive power for finding severity, albeit with a noisy relationship to the assessed rankings.

### 3.3. Comparison of a binary classifier with RankNet for age ranking.

In Fig.3 we compare age ranking performance for binary classifier models with RankNet models trained on equal sized training sets of varying sizes. Performance on all metrics is stronger ($p < 0.001$) using RankNet as compared with the binary model, reaching ME of 7.126 years for the binary classifier versus 5.653 years for RankNet with $1,000,000$ training images. This is unsurprising given the much higher level of supervision being provided with the ranked training data. More interestingly we can approximately quantify the reduction

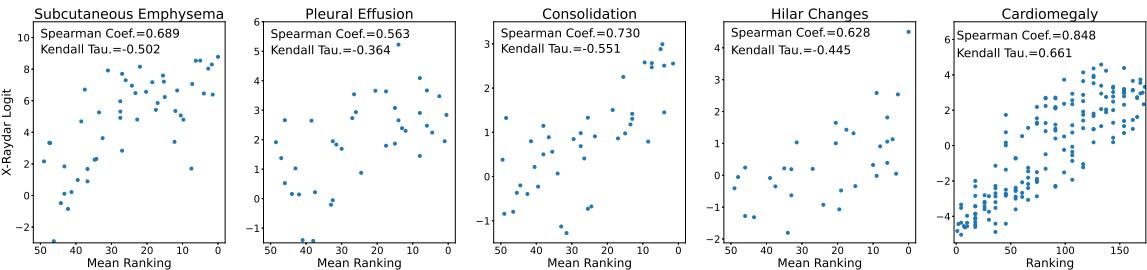

Figure 2: Binary classifier logits outputs for each named finding versus the mean severity ranking for each test set. Positive ranking coefficients are found for all findings indicating that the magnitude of the model output is predictive of finding severity.

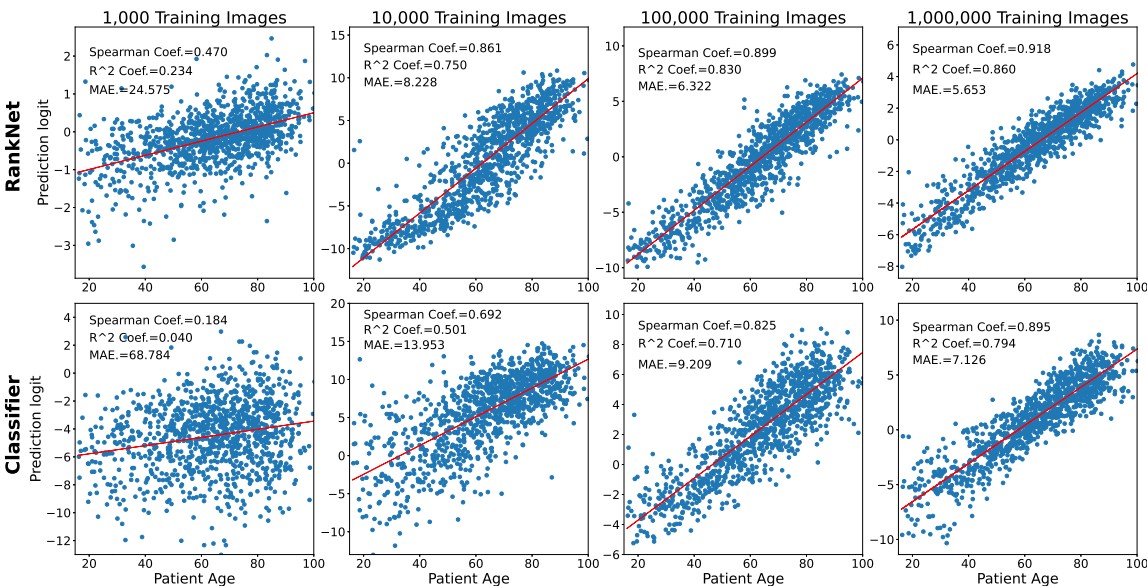

Figure 3: Comparisons of models trained on training sets of increasing size, using (top) RankNet with the training images given ranking labels in each minibatch, and (bottom) a binary classifier trained with BCE loss with the training data assigned only binary labels. Ranking metrics assessed on the test set are unsurprisingly stronger for RankNet in each case.

in training set size required with fully ranked training data to give equivalent performance to binary supervision; as a coarse estimate, interpolating between the data points would indicate similar ranking performance using 10,000 ranked training images with RankNet and 200,000 training images with the binary classifier.

Two main implications for severity ranking can be inferred from this experiment. Firstly, even with ranked data a relatively large amount of images are required to give a robust severity measure; in the case of patient age, approximately 10,000 ground truth ranked images are required to materially improve age prediction above the naive estimation MAE of 8.70 years where the mean patient age is estimated for all images in each category. Fully hand ranking this quantity of images is impractical. Secondly, a large quantity of binary labelled data can go some distance to emulating a smaller ground truth ranked dataset; we estimate that the full binary labelled training set gives a similar outcome to 20,000 fully ranked images which supports the idea that binary labels can yield a useful ranking measure.

### 3.4. Effect of including semi-supervised ranking information during training

Table 1: Comparison of age prediction MAE between binary classifier logit, RankNet, and combined loss models using 1,000, 10,000, and 100,000 ranked data points for ranking loss. The combined model outperforms RankNet in all instances, and outperforms the baseline classifier with 10k and 100k ranked images ($p < 0.001$). 95% confidence intervals are quoted based on 1,000 bootstrapping samples. Best performing model for each ranked data size marked in **bold**.

| | Ranked Data Points | | |
| Model | 1,000 | 10,000 | 100,000 |
| --- | --- | --- | --- |
| Classifier | **7.126** (7.090-7.160) | 7.126 (7.090-7.160) | 7.126 (7.090-7.160) |
| RankNet | 24.574 (24.441-24.699) | 8.228 (8.191-8.264) | 6.322 (6.291-6.353) |
| Combined | 7.257 (7.223-7.290) | **6.607** (6.577-6.637) | **6.145** (6.115-6.174) |

Finally, we investigate whether ranking performance can be improved in a hybrid semi-supervised approach, leveraging the full training set for binary classification along with varying numbers of ranked images in the RankNet training subset. We compare the age ranking performance of (a) a binary classifier trained with the full binary labelled training set, (b) RankNet using only ranked training sets of 1,000, 10,000 and 100,000 images, and (c) a combined loss model using both the full binary labelled training set and the ranked subset. The results of this experiment are summarised in Table 1 (with scatter plots shown in the appendix Fig.7). We observe that the combined loss model outperforms RankNet for all sizes of ranked training subsets, and outperforms the baseline classifier (MAE 7.126 years) with 10,000 (MAE 6.608 years) and 100,000 (MAE 6.145 years) ranked images, although the baseline classifier slightly outperforms the combined model with only 1,000 ranked images added (MAE 7.257 years).

From this experiment we conclude that combining ranked and binary data can lead to improved model performance in excess of either approach in isolation. There also appears to be a threshold amount of ranked data required to boost performance; it is likely that when only a small amount of ranked data is used for the ranking loss, our approach overfits to this subset and degrades overall performance. It is possible that this could be addressed with changes to the training process to reduce the number of batches being fed to the RankNet loss per epoch, as in the current implementation the ranked data is repeatedly cycled over

during each epoch of binary classifier training. In any case, in the current set-up we can estimate the number of ranked training images required to give improved performance over the classifier baseline by interpolating between the the experiments performed. This yields an estimate of 1,800 ranked images above which there is improvement to the overall model performance. While this is a substantial amount of ranked data, it is well within the range of a practical number of images for human experts to annotate for real clinical findings, and may be possible to apply to real clinical findings in future.

## 4. Conclusion

In this paper we have explored abnormality severity ranking in chest x-rays in the binary label and semi-supervised contexts. We first considered the degree to which nuanced intra-finding severity information can be extracted from a classifier model pre-trained on data with only binary abnormality labelling. The pre-sigmoid raw output from an open-source chest x-ray classifier, X-Raydar, was found to be predictive of the severity rankings produced by two radiologists on test sets of images for four radiological findings, with Spearman's coefficients from 0.563 (pleural effusion, N=41) to 0.730 (consolidation, N=41), and the cardiomegaly output was predictive of measured cardiothoracic ratio ranking (0.848, N=173).

We then examined the relative value of binary labelled versus fully ranked training data in approaching the severity ranking problem, using patient age as a proxy for a radiological finding for which a full ground truth ranking is available. Comparing ranking models trained with binary versus fully ranked training data and corresponding objective functions, we were able to estimate the amount of binary auto-labelled training data which gave similar ranking performance to a fully supervised ranking model. We found approximately a 20 times increase in training data was required to give similar ranking performance using a binary classifier model. This illustrates the benefit of fully annotated data in the training process, but also demonstrates that a sufficiently large quantity of binary labels can potentially give similar benefit to a relatively large amount of resource-intensive annotated data.

Finally we explored combining binary and fully ranked training data to to train a single age prediction model with a hybrid objective function. Adding a moderate amount of ranked data to the much larger set of binary labelled data was found to boost regression and ranking performance above the baseline binary classifier level. This indicates that a semi-supervised approach supplementing hand annotated labels with larger volumes of auto-labelled images could offer improvements in model performance versus either approach in isolation, which could be beneficially in optimising clinician time required for manual annotations in future machine learning studies.

Approaching severity estimation in a full ranking setting gave valuable insights into both model performance and inter-operator ranking consistency. However, the additional time required to produce ranked test sets as opposed to using a graded severity bucket approach limited both the test set sizes and the number of findings we were able to evaluate with the available resources. To build on this work, incorporating additional radiologists' rankings would allow a more detailed analysis of inter-operator consistency and allow for more robust consensus rankings to be obtained to strengthen our conclusions.

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

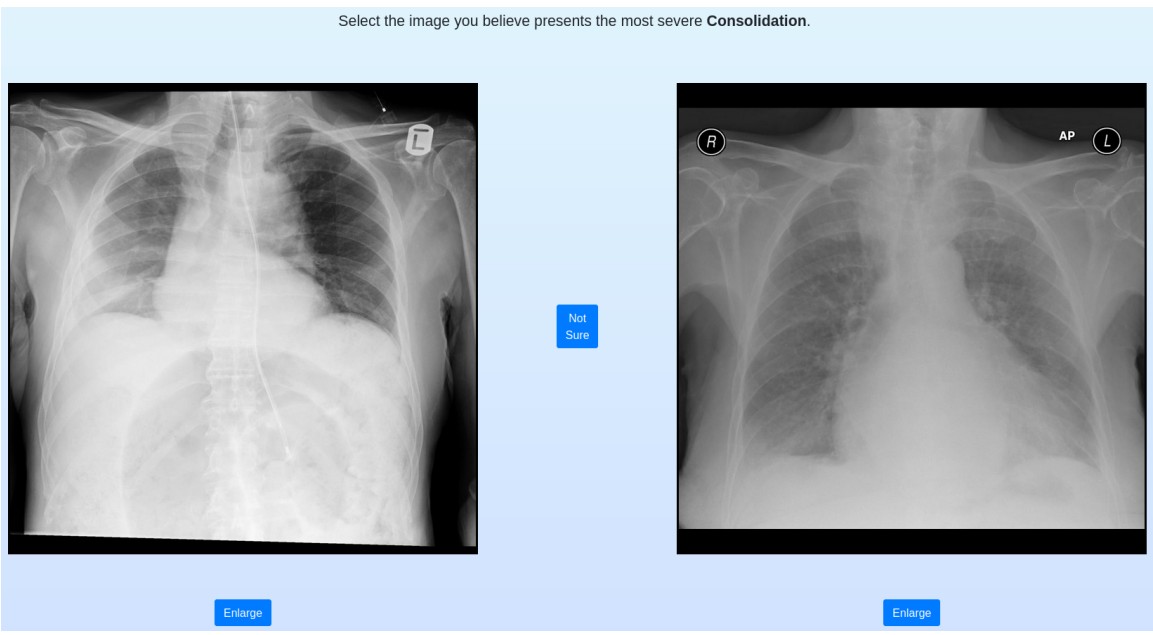

Figure 4: Screenshot of the interface used for pairwise ranking of the test set images. The radiologist selected which of the images represented a more severe example of the named finding. New pairs were selected online by the merge sort algorithm, and the radiologist ranked each pair presented until the algorithm was completed.

## Appendix A. Test set ranking interface

In Figure 4 we show an example of the web interface used by the radiologists to rank the test sets for each of four findings. For each set, 50 images were sampled from those labelled positively with the named finding (and potentially with comorbidities) from our hand labelled ground truth image set. The merge sort algorithm was used to present the radiologist with successive pairs of images in png format at 1024x1024 resolution, and the image deemed to show the greatest severity was selected. Successive pairs were assessed until the algorithm was completed. Figure 5 shows examples of increasing ranked severity for each of the findings alogn with cardiomegaly, which was assessed by manual measurement of the cardiothoracic ratio.

## Appendix B. Model training parameters

Our age binary classifer and ranking models were based on the EfficientNet-B3 CNN backbone initialised with ImageNet pre-trained weights and with a single neuron output after the 1536 dimension feature map layer. An ADAM optimizer with initial learning rate 0.001 and 1e-4 weight decay factor, adaptive learning rate decay with a 0.5 decay factor and a three epoch patience level was used. We report test results for the lowest validation set error epoch in each case. Models were trained on all 8 V100 GPUs of an Nvidia DGX1

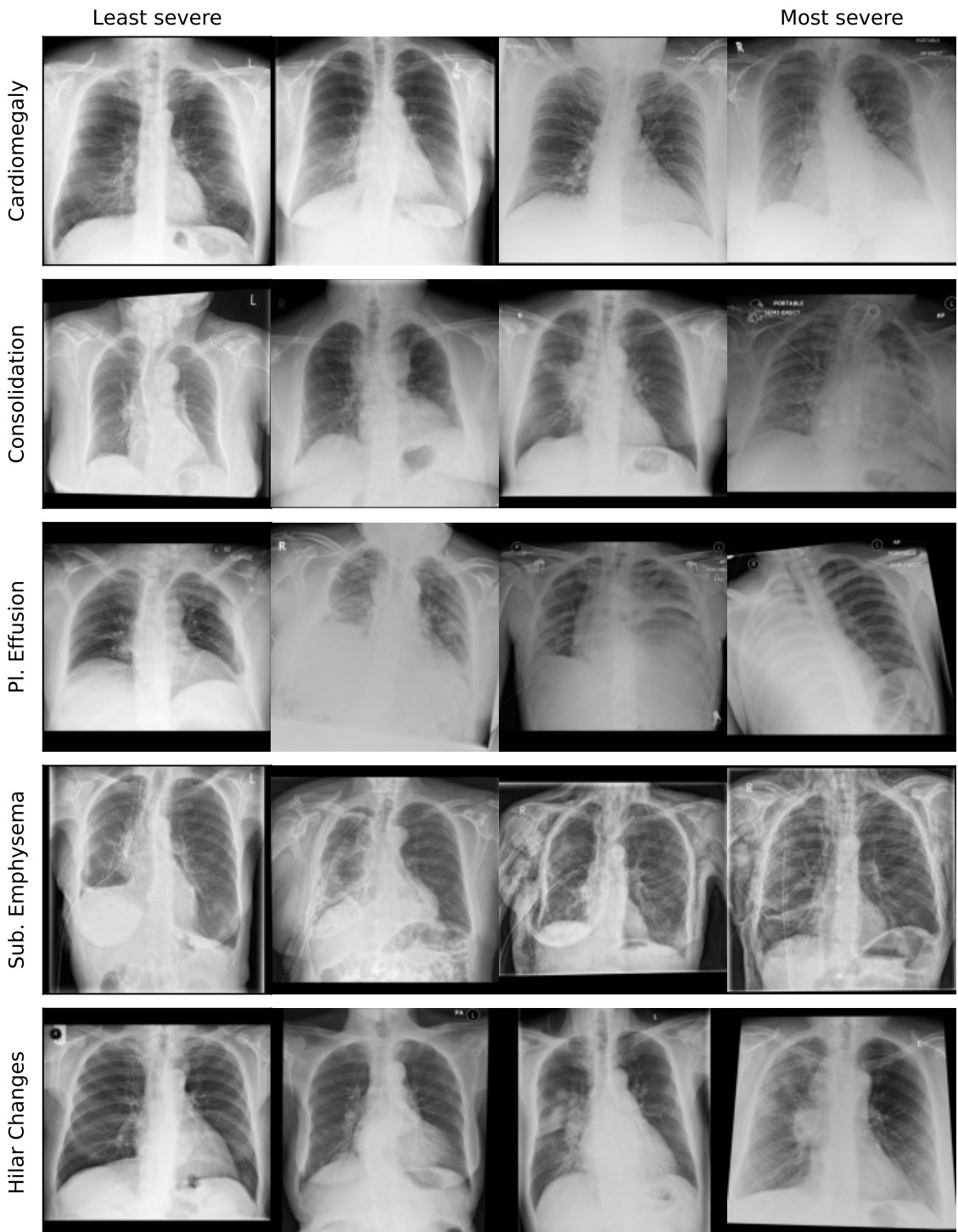

Figure 5: Examples of radiological findings of increasing mean ranked severity for each of the radiological findings considered. Cardiomegaly is ranked by measured cardiothoracic ratio, while the other findings are ranked by the pairwise comparison method.

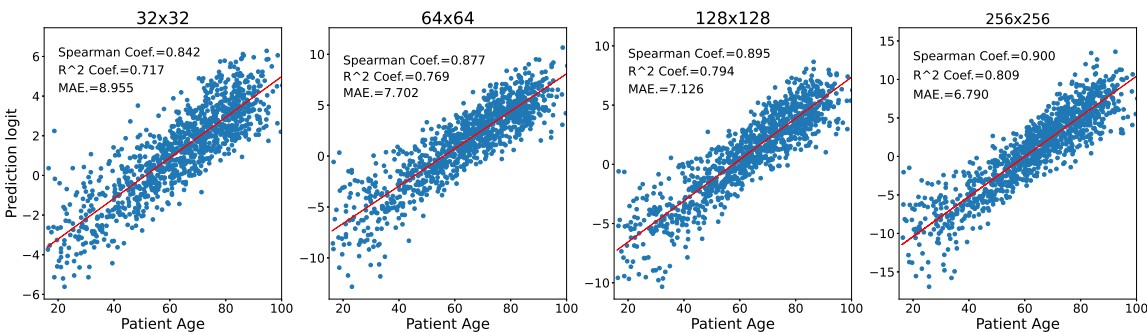

Figure 6: Plots of logits outputs for patient age binary classifier versus true patient age, for models trained at four input resolutions. Plots are shown for all test images (top) and only for those test images classified as 'old' (i.e. > 55 years) to act as a proxy clinical finding. Spearman's rank coefficient and true age prediction with a linear regression model are seen to improve with model resolution.

server at 256x256 resolution, with batch sizes of 512 for the binary classifier loss batches and 200 for the RankNet batches. In the semi-supervised approach (equation 3), the value of the hyperparameter $\lambda$ was established by a manual parameter sweep to approximately match the magnitude of the two loss components.

## Appendix C. Impact of image resolution on model ranking performance

In Figure 6 we show scatters of model logits output versus true patient age using binary age classification models (i.e. young versus old with respect to a 55 years cut-off) trained at four increasing resolutions. Ranking and age regression performance is seen to improve consistently with higher model resolution. This is not entirely intuitive, since improved binary classification performance is not necessarily linked to stronger linear regression performance (e.g. the case of a binary step function around 55 years which would classify perfectly but rank poorly).

## Appendix D. Comparative scatter plots of binary classifier, RankNet and combined approach age models

In Figure 7 we show scatter plots of model logits output versus true patient age for binary classifier, RankNet and combined model approaches under different data conditions. In all cases the full set of binary-labelled training images was available to the model (although not used by the RankNet model). In addition, from 1,000 to 100,000 images were given full ground truth rankings in each case (although these annotations were not used by the classifier model). Above 1,000 ranked training images, the combined model using both classifier and ranking objective function was found to give the strongest ranking and regression performance as assessed with Spearman coefficient, $R^2$, and MAE for true age regression.

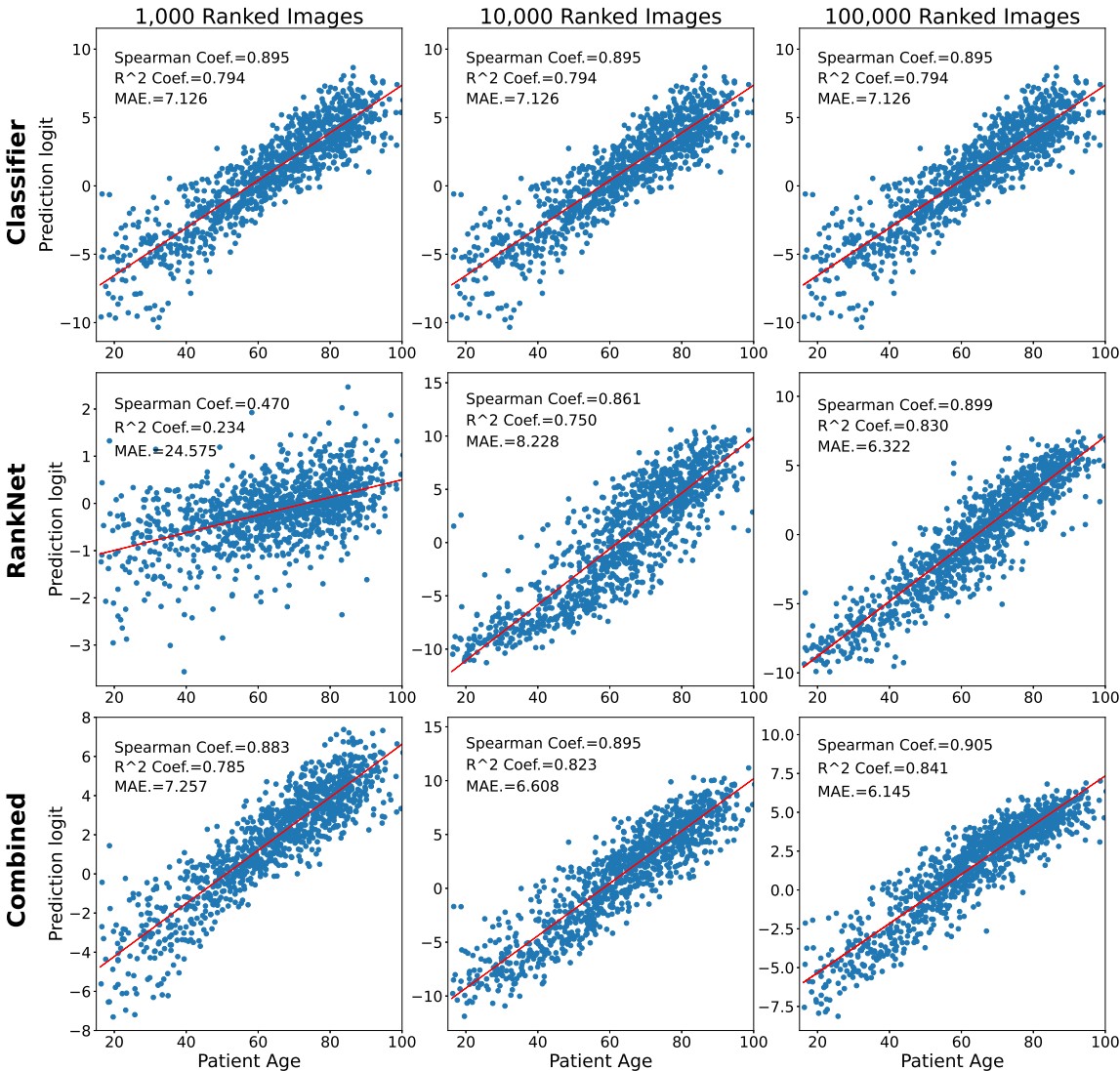

Figure 7: Comparison of a binary classifier logit output, RankNet output and combined loss model for ranking performance of patient age. The classifier is trained with the full binary label set without ranked labels in each case (hence the output is invariant to the ranked data count), while RankNet is trained with only the ranked subset specified in the column headings. The combined loss model, trained with both the full binary set and ranked subset, outperforms both the independent classifier and RankNet models with 10,000 and 100,000 ranked training images, implying that incorporating ranked training data can improve classifier performance.

