# OpenReview forum: "Automated ranking of chest x-ray radiological finding severity in a binary label setting"
_MIDL.io/2024/Conference — MIDL 2024 Poster_

### Official Review · Reviewer_9jNV · 2024-02-20

**Confidence:** 3
**Preliminary Rating:** 2
**Recommendation:** Poster
**Final Rating:** 2

**Summary:**

The paper proposes to study the correlation of network outputs, trained for multi-label disease detection with binary outputs in chest X-ray images, to disease severity. The authors show a positive correlation. In addition, they include explicit ranking annotations. and propose using RankLoss to embed this information during training, which shows with correlation with patient aging. Finally, they explore semi-supervised strategies to diminish the amount of necessary ranked labels to calibrate the network.

**Strengths:**

- The observation of using output logits to evaluate severity, in the context of chest X-ray, is original.
- The results show a positive correlation between disease severity and age, in most of the cases.

**Weaknesses:**

- The objective of the work, i.e. exploring the capability of a trained network to grade severity in findings/patients, is highly related to the problem of network calibration [a]. Nevertheless, the authors explore this issue through an experimental scope. A more methodological perspective, based on optimization basis, would be expected.
- Indeed, the rank loss explicitly calibrates the outputs, based on the ordering via annotated confidence of severity,  which then provides better metrics in the evaluated tasks, but requires an annotation process.
- Following the previous comment, several baselines on network calibration could be incorporated into the work to complement the Classifier baseline: label smoothing [b], focal loss [c], or logit penalties [d], for example.
- For instance, from Figure 3, comparing RankNet and Classifier methods, RankNet usually obtains an overall smaller logit absolute value scale (I would want to see concrete values for this, but it seems the case for most of the cases), which is known to be correlated with good calibration [d]. This observation is aligned with the better correlation of such logits to severity (in the form of patient age in this case).
- There are some experimental decisions that I do not fully understand: why the correlation to severity predictions (Section 3.2.) is only evaluated using X-Raydar, and not the same methods used for age ranking (e.g. so-called classifier and RankNet)?
- Finally, some experimental details are missing: train/val/test partition, or how the lambda hyperparameter on Eq. 3 is fixed.

[a] On the calibration of moder neural networks, ICML, 2017.

[b] Rethinking the inception architecture for computer vision, CVPR, 2016.

[c] Focal loss for dense object detection, CVPR, 2017

[d] The devil is in the margin: Margin-based label smoothing for network calibration, CVPR. 2022.

**Detailed Comments:**

No additional comments.

**Justification Of Final Rating:**

After the authors' rebuttal, I based my evaluation on their main contributions, that: "focuses on utilizing the logit outputs from models trained exclusively on binary labels to infer a nuanced spectrum of severity for radiological findings", and that "The rank loss discussion in our context is hypothetical and aimed at future research directions rather than a description of our current methodology". Then, the main contribution of this work is evaluating the previous authors' model, X-Raydar, trained on binary labels, to severity ranking (Figure 2). Further experiments using RankLoss and Classifier to evaluate age ranking do not provide a direct evaluation of the correlation to severity, and authors claim it to be a hypothetical, future research direction, to predict disease severity. Based on such observation, in my opinion, the paper does not provide enough content for its acceptance as a full conference paper. I justify this claim by the absence of methodological grounding or technical novelty, and the limited experiments on the target setting: disease severity ranking.

**Justification Of The Preliminary Rating:**

The work is largely experimental, and the drawn conclusions might be too specific to the proposed experimental setting. My recommendation is based on the absence of a methodologically grounded construction of the severity ranking problem, and a few experimental inconsistencies and missing details.

**Questions To Address In The Rebuttal:**

Please, see Weaknesses.

**Special Issue:**

No

---

> ### Author Response · Authors · 2024-03-17
> **Responses to reviewer 9jNV**
>
> Reviewer 9jNV
> Strengths:
> We are glad that you found our observation of using output logits to evaluate severity in chest X-rays to be original. This was a key motivation for our study and thank you for noting the positive correlation between disease severity and age in most cases.
>
> 1/ The objective of the work, i.e. exploring the capability of a trained network to grade severity in findings/patients, is highly related to the problem of network calibration [a]. Nevertheless, the authors explore this issue through an experimental scope. A more methodological perspective, based on optimization basis, would be expected.
>
> - Thank you for your feedback and for highlighting the connection with network calibration. However, it's important to clarify the unique context and objectives of our study, which may have contributed to a misunderstanding regarding the applicability of traditional calibration methods here.
> Our research focuses on utilizing the logit outputs from models trained exclusively on binary labels to infer a nuanced spectrum of severity for radiological findings. This approach diverges from conventional scenarios where calibration is typically applied to adjust the confidence levels of binary or multi-class classification outputs to match observed frequencies.
> The novel aspect of our work lies in the exploration of whether these binary-trained models, without modification or the direct application of calibration techniques, can inherently provide insights into severity gradations. This investigation is fundamentally experimental and aims to unearth new applications and interpretations of model outputs beyond their initial binary classification purpose.
> We believe that the demonstration of a correlation between model logits and expert-derived severity rankings—achieved without the explicit introduction of calibration techniques—serves as a foundational step towards understanding the potential of existing models to offer clinical value in severity estimation tasks.
> We want to make clear that our training used only binary labels (just indicating if an abnormality was present or not) without any information about how severe the abnormality was, as this is not routinely assessed by the radiologists. But for testing, we didn't use any binary labels. Instead, we had radiologists rank the images based on how severe the condition looked, using a method where they compared many pairs of images. Our main goal was to see if the outputs from our model, which was only trained on binary labels, could still guess the right order of severity in these ranked images. We've made some changes to the manuscript to better explain this motivation and unique setup, hoping to clear up any confusion about how we used calibration methods in our study. This should help show how our approach is different and why traditional calibration methods might not fit exactly with what we were trying to do.
>
> 2/ Indeed, the rank loss explicitly calibrates the outputs, based on the ordering via annotated confidence of severity, which then provides better metrics in the evaluated tasks, but requires an annotation process.
>
> - Thank you for highlighting the role of rank loss in calibration. Again here, it's important to clarify that our study does not use severity annotations for training the models. Instead, we explore the potential of binary-trained models to infer severity rankings based on their output logits. The rank loss discussion in our context is hypothetical and aimed at future research directions rather than a description of our current methodology. Our approach leverages binary labels for training, and the inference about severity is drawn post hoc from model outputs, without the necessity for additional severity-specific annotations.

---

> > ### Author Response · Authors · 2024-03-17
> > **Responses to reviewer 9jNV (cont)**
> >
> > 3/ Following the previous comment, several baselines on network calibration could be incorporated into the work to complement the Classifier baseline: label smoothing [b], focal loss [c], or logit penalties [d], for example.
> >
> > - For this study, we decided not to use calibration methods on purpose. Our main goal was to see if models that are only trained to say if an abnormality is present or not can still guess how severe an abnormality is. We're really focusing on what these models can do on their own, without changing them or using special techniques to adjust their outputs. Bringing in calibration techniques now would change the direction of this initial work. It might make it harder to see the main point we're trying to make: that these simple models might already be able to give us useful information about severity, just as they are. We think this is an important idea to explore all by itself, without making things more complicated at first.
> > That said, we know calibration methods are useful and might help make severity predictions even better in the future. We plan to look into these techniques in our next steps, building on what we've learned from this initial exploration.
> > For instance, from Figure 3, comparing RankNet and Classifier methods, RankNet usually obtains an overall smaller logit absolute value scale (I would want to see concrete values for this, but it seems the case for most of the cases), which is known to be correlated with good calibration [d]. This observation is aligned with the better correlation of such logits to severity (in the form of patient age in this case).
> > Your observation regarding the logit scale differences between RankNet and the classifier method offers a valuable perspective on model calibration, highlighting RankNet's smaller logit scale that might suggest better calibration, potentially contributing to its effectiveness in severity ranking as shown in Figure 3. However, it's crucial to reemphasize the core intention of our investigation within this broader discussion. Our study's primary aim is to explore the capability of binary-trained models to infer severity rankings directly from their output logits, without leveraging severity annotations for training. This approach is deliberately chosen to evaluate the inherent potential of these models to extend beyond their original binary classification purpose, venturing into the realm of severity estimation in a clinical context.
> > Given this unique premise, the observed differences in logit scale and the subsequent implications for model calibration are intriguing but secondary to our main objective. While traditionally, smaller logit scales are associated with better-calibrated models, our focus remains squarely on demonstrating how unaltered outputs from binary-trained models can provide valuable insights into severity rankings. This exploration seeks to challenge and expand the conventional applications of machine learning models in medical imaging, without initially delving into optimization or calibration based on severity scales Thus, while we acknowledge the validity and potential importance of your points for a more traditional calibration study, they fall outside the direct scope of this initial experimental study.
> >
> > 4/ There are some experimental decisions that I do not fully understand: why the correlation to severity predictions (Section 3.2.) is only evaluated using X-Raydar, and not the same methods used for age ranking (e.g. so-called classifier and RankNet)?
> >
> > - To clarify this point, in Section 3.2, we used only the pre-trained X-Raydar model for severity prediction on radiologist-ranked findings, while in Section 3.3, we compared our custom classifier and RankNet models on the age ranking task. The primary reason for this difference was the nature of the ground truth data available in each case. For the radiological findings, we had subjective rankings from radiologists, which were more suited for evaluation with the pre-trained X-Raydar model that was specifically designed for these findings. In contrast, for age prediction, we had exact ages available, allowing us to train and compare the performance of our custom classifier and RankNet models. We agree that evaluating the classifier and RankNet models on the radiologist-ranked findings alongside X-Raydar would provide a more direct and consistent comparison across the different severity prediction tasks. While we focused on X-Raydar for the radiological findings in this study due to its strong performance on this specific task in our previous work, we believe that extending the evaluation to include the classifier and RankNet models is a valuable direction for future research. This would help to provide a more comprehensive picture of the relative strengths and weaknesses of different model architectures for severity prediction across a range of tasks and ground truth label types.

---

> > > ### Author Response · Authors · 2024-03-17
> > > **Responses to reviewer 9jNV (cont 2)**
> > >
> > > 5/ Finally, some experimental details are missing: train/val/test partition, or how the lambda hyperparameter on Eq. 3 is fixed.
> > >
> > > - These omissions have been corrected in the methods and appendix of the revised manuscript. Details of the training splits are provided in our earlier work (Cid et al. 2024, Development and validation of open-source deep neural networks for comprehensive chest x-ray reading: a retrospective, multicentre study. The Lancet Digital Health) as referenced on page 3 but could be detailed in an appendix section if that is insufficient.

---

> > > > ### Comment · Reviewer_9jNV · 2024-03-23
> > > >
> > > > I would want to thank the authors for their detailed clarifications and comments.

---

### Official Review · Reviewer_NYud · 2024-02-26

**Confidence:** 3
**Preliminary Rating:** 2
**Final Rating:** 2

**Summary:**

This work examines the ability of models trained with only binary training data to give abnormality severity information from their raw outputs. They assess performance on a ranking task using manually ranked test sets for each of the five findings. The pre-sigmoid raw output from an open-source chest x-ray classifier predicted the severity rankings produced by two radiologists on test sets of images for four radiological findings. Then they examined the relative value of binary labeled versus fully ranked training data in approaching the severity ranking problem, using patient age as a proxy for a radiological finding for which a full ground truth ranking is available.

**Strengths:**

1. This work invites radiologists to examine diseases in CXR images, which have a certain degree of reliability.
2. Considering the severity of a disease is a clinically relevant idea that will help radiologists diagnose a patient's condition.

**Weaknesses:**

1. The design of disease severity could be more clear, and more details of the relationship between disease severity and label could be provided.
2. The purpose of using age as a feature is not clear.
3. Lack of comparison on public datasets.

**Detailed Comments:**

1. How accurate is CTR alone in diagnosing cardiomegaly? Some reasons may prevent CTR from accurately diagnosing. For example, CXR images have AP and PA orientations. The different orientations may cause the same disease severity having slightly different CTRs.
2. Page 3 says: "After eliminations, we retained 173 ranked images for cardiomegaly, 41 each for pleural effusion and consolidation, 50 for subcutaneous emphysema and 32 for paratracheal hilar changes. The retained images were scored with the mean ranking of the two radiologists to reach a final ranking measure." How many severity levels are there for each disease?
3. Page 3 says: "Patient age was used as a ’toy’ proxy for a radiological finding with ground truth ranking available via the chronological patient age.". What does this mean? Do you mean the disease severity has a relationship with age? The Fig.3 results seem to demonstrate the relationship between disease severity and age. However, the severity of a disease in a patient may vary greatly over time. Is there any practical value in exploring this relationship?
4. Eq. 3 says the total loss is the sum of BCE loss and rank loss, and Eq. 1 represents the BCE loss, please describe rank loss $L_{Ranking}$ in detail. How to set the balance hyperparameter $\lambda$?
5. How to process the corresponding labels for diseases of different severities? Are they all 0-1 labels? Given two CXR images, one is with the most severe effusion, the other is with moderate severity, What are the label values of effusion for the two images? Are the prediction logits output by the model related to the predicted disease severity? Please provide more details.
6. How does the model perform on publicly available data sets? For example MIMIC CXR dataset?

**Justification Of Final Rating:**

After the response given by the author, I think the contribution of this paper is the use of binary logits to assess disease severity. In this idea, the author uses loss_RankNet as the loss function of paired images to evaluate the difference in severity. This idea seems reasonable and there are many experiments. However, the paper lacks both a quantification of severity and an explanation of the phenomena. Based on these observations I give the opinion of weak rejection.

**Justification Of The Preliminary Rating:**

1. This work designs a process to rank the disease, and the experiments seem to be reasonable.
2. The lack of detail about disease severity makes it difficult to concretely relate it to the model’s label values.
3. Experimental results without publicly available datasets reduce credibility.

**Questions To Address In The Rebuttal:**

1. Detailed introduction to the settings of disease severity, the relationship between severity and label value, and the relationship between severity and predict logits.
2. More details about the $L_{Ranking}$.
3. Experiments on public datasets would be welcome.

---

> ### Author Response · Authors · 2024-03-17
> **Responses to reviewer NYud**
>
> 1. How accurate is CTR alone in diagnosing cardiomegaly?...
>
> - Thanks for raising this point. The PA/AP effect on the measured vs ‘true’ CTR was not considered for this test set, which uses a mix of positionings. It is likely that making a consistent correction for AP images would somewhat improve the quality of the ranking performance, although at the same time it isn’t obvious if the pre-trained classifier has learnt an implicit adjustment in its output for the two cases, given that the model was pre-trained on comingled AP/PA films.
>
> 2. Page 3 says: "After eliminations, we retained 173 ranked images for cardiomegaly... " How many severity levels are there for each disease?
>
> 3. Page 3 says: "Patient age was used as a ’toy’ proxy for a radiological finding with ground truth ranking available via the chronological patient age.". What does this mean?...
>
> - We apologize for the confusion in our description of using patient age as a proxy for a radiological finding. In this section, we explored the relationship between a model's predicted age and the ground truth patient age, completely separate from any disease presence or severity. The purpose was to use age as a toy example of a variable with a known ground truth ranking, to allow us to compare the performance of different modeling approaches (binary classifier vs. RankNet) in a setting where we have a definitive ground truth ordering.
> Specifically, we trained a binary classifier to predict whether a patient is 'young' or 'old' based on a threshold of 55 years. We then evaluated how well the model's raw output (logit) for the 'old' class correlated with the patient's actual age. This is analogous to training a binary classifier to predict the presence or absence of a radiological finding, and then evaluating how well its raw output correlates with the finding's underlying severity.
> Figure 3 visualizes this relationship by plotting the model's output against the true patient age. It does not depict any relationship between age and disease severity. The intent was to use age as a stand-in for disease severity, since age provides a clear, indisputable ranking (older patients are always "more severe" than younger patients in this toy example).
> We agree that in a real clinical setting, the severity of a disease may vary greatly over time and is not necessarily correlated with age. The practical value of this experiment was not to suggest that age is a useful proxy for disease severity, but rather to provide a controlled setting in which to compare the ability of different modeling approaches to capture an underlying severity ranking, using a variable (age) for which we have a perfect ground truth ordering. The insights gained from this experiment can inform the design of models for predicting the severity of actual radiological findings, even though age itself would not be a useful predictor in that context.
> We have revised the description of this experiment in the paper to clarify its purpose and avoid any confusion about the relationship between age and disease severity.

---

> > ### Author Response · Authors · 2024-03-17
> > **Responses to reviewer NYud (cont).**
> >
> > 4. Eq. 3 says the total loss is the sum of BCE loss and rank loss, and Eq. 1 represents the BCE loss, please describe rank loss in detail. How to set the balance hyperparameter lambda?
> >
> > - As per the response to reviewer tMQD (point 2), the notation here was inconsistent and has been corrected. The RankNet loss should have been referenced in Equation 3, and is fully described in Equations 1 and 2. The balance parameter lambda was set by a manual parameter sweep to approximately equalize the magnitude of the two loss components; this has been added to Appendix B.
> >
> > 5. How to process the corresponding labels for diseases of different severities? Are they all 0-1 labels? Given two CXR images, one is with the most severe effusion, the other is with moderate severity, What are the label values of effusion for the two images? Are the prediction logits output by the model related to the predicted disease severity? Please provide more details.
> >
> > - The question regarding labels for different disease severities appears to stem from the misunderstanding discussed in point 2. To clarify, the classifier models in our study were trained using only binary labels (0 for absence, 1 for presence) derived from clinical reports using natural language processing (NLP). No severity information was used during the training process.
> > Severity rankings were used solely to construct the test sets for evaluating the models' ability to predict relative severity. In these test sets, the images were fully ranked from least to most severe using the pairwise comparison approach described in the response to point 2. There were no label values assigned to the images in the test sets in the sense of 0-1 or severity buckets.
> > Given two CXR images in a test set, one with the most severe effusion and the other with moderate severity, both images would have a binary label of 1 for the presence of effusion. However, they would have different ranks in the severity ordering produced by the radiologists. The goal of our study was to assess how well the prediction logits of the models, trained using only binary labels, could predict this severity ranking.
> > So in summary:
> > -	Training labels: Binary 0-1 only, no severity information
> > -	Test set labels: No 0-1 labels, images were manually ranked for severity by radiologists using pairwise comparisons
> > -	Model predictions: Logit outputs, which we aimed to show were predictive of severity rankings despite the model being trained with only binary labels
> > Our paper does not claim to predict absolute severity levels, but rather relative severity rankings. The key finding is that the prediction logits of models trained with only binary labels can nonetheless be predictive of these severity rankings, suggesting that some implicit severity information is captured in the model's confidence scores.
> > We hope this clarifies the labeling and evaluation process used in our study. We have improved the explanation in the paper.
> >
> > 6. How does the model perform on publicly available data sets? For example MIMIC CXR dataset?
> >
> > - Unfortunately, we currently lack funding for additional clinician time to produce ordered test sets on MIMIC-CXR, but producing a public benchmark test set for ranking algorithms would be a valuable contribution. We are speaking with our data providers about making available the test images used in a public repo, which would require an exemption under our existing data sharing agreement. Our prior work on e.g. the X-Raydar model indicates that our data generalises well to other DICOM-available datasets, and reproducing the results there will be a priority when resources allow.

---

> > > ### Comment · Reviewer_NYud · 2024-03-26
> > >
> > > Thanks for your reply and clarification. I think the feedback clarifies my doubts, I have no further questions.

---

### Official Review · Reviewer_tMQD · 2024-03-02

**Confidence:** 3
**Preliminary Rating:** 4
**Recommendation:** Poster
**Final Rating:** 4

**Summary:**

this is an interesting paper with the aim of examining the ability of models trained with only binary training data to give useful abnormality severity information from their raw outputs. Cardiomegaly, consolidation, paratracheal hilar changes, pleural effusion and subcutaneous emphysema are the ones pathologies in the x-ray, and authors tried to rank them with their binary models. Authors correlated the findings with automated vs human ranking and obtained from 0.5 to 0.8 Pearson correlation. Later there is a comparison with RankNet with semi-supervised setting, and improved performance is obtained compared to initially defined model.

**Strengths:**

-Evaluating the continuous logits outputs of a chest x-ray multi-label classifier, trained only on NLP-generated binary labels, as a predictor of severity ranking across a range of radiological findings.
-assumption that the level of certainty of a binary model is likely to correlate with the extent or obviousness of the pre- sentation of the finding in the image sounds solid and true.
- Involving two radiologists to correlate their findings with the ranking by the algorithm is less biased than single radiologist.
- useful for scaling the model to large scale models.
- results are promising.

**Weaknesses:**

- the idea is incrementally novel, but there is technically almost no innovation.
-lamda_rankinvg is. not available in the equations, I guess there is a typo. it should be LambdaRankNet, I assume.
-pleural effusion is not a superb difficult pathology to rank, any idea why the results are poorer?

**Detailed Comments:**

my comments are pretty much same and self-contained for weaknesses and positive sides.

**Justification Of Final Rating:**

I think this is a fair conference paper with some innovations, but not so strong in novelties.
The response were satisfactory to some extent,
due to lack of novelty and coherence problems in the paper, I keep my score at weak accept for a fair conference acceptance, as this paper is in between strong accept and weak accept, to my opinion.

**Justification Of The Preliminary Rating:**

I think the idea is nice, and practical. Innovation may be not there but it sounds like a fair conference article with a well  defined hypothesis and authors tested the test with x-ray test with two radiologists involved. Nicely written easy to follow paper with promising results.

**Questions To Address In The Rebuttal:**

1) can authors have a subsection and clearly mention what is new in this paper? the idea is yes, but not technical.
2) there are some formulations wrong, I assume, not clear if lamda_ranking is ever defined.
3) why some pathologies are having higher uncertainties while their visibility is easier (pleural effusions).
4) are there any other uncertainty algorithm that can be applied to reduce the gap and increase the severity estimation?

**Special Issue:**

No

---

> ### Author Response · Authors · 2024-03-17
> **Responses to reviewer tMQD**
>
> Reviewer tMQD
> 1. Can authors have a subsection and clearly mention what is new in this paper? the idea is yes, but not technical.
>
> -The final paragraph of the introduction has been redrafted into a bulleted contributions list to make clearer the intended paper contributions.
>
> 2. There are some formulations wrong, I assume, not clear if lamda_ranking is ever defined.
>
> - The formulations in this methods section have been corrected to be consistent between the text and equations, and equation 3 amended slightly to show lambda as a separate hyperparameter preceding the ranking loss.
>
> 3. Why some pathologies are having higher uncertainties while their visibility is easier (pleural effusions).
>
> -This is an interesting point that we didn't have space to fully discuss in the paper. Inter-operator ranking agreement was actually weakest for pleural effusion among the findings considered, indicating that objectively ranking this finding may have been more challenging or controversial for the clinicians. Recruiting additional radiologists to repeat the ranking could help shed light on this. Additionally, if the model is generally confident in its pleural effusion predictions, the severity range over which uncertainty is informative may be smaller. Investigating these factors would be interesting future work to better understand the performance variation across findings.
>
> 4. Are there any other uncertainty algorithm that can be applied to reduce the gap and increase the severity estimation?
>
> - Exploring alternative uncertainty quantification approaches is a great suggestion for potentially improving severity estimation. Techniques like model ensembling, test-time augmentation, or Bayesian approximation could be interesting to investigate. However, a thorough comparison of uncertainty methods was beyond the scope of this initial proof-of-concept study. We see this as a valuable direction for future work building upon our findings.

---

> > ### Comment · Reviewer_tMQD · 2024-03-23
> >
> > thank you for the comments.

---

### Author Response · Authors · 2024-03-17
**Response to reviewers**

We thank the three reviewers for their detailed comments, in response to which we have drafted a revised manuscript. Below, we first provide a general response to the reviewers, followed by point-by-point responses to each reviewer referencing the relevant changes made where relevant.

Overall, the reviewers recognise that the subject matter (i.e. assessing finding severity from weakly supervised data) has novelty and is of clinical interest but highlight the lack of technical innovation in the work. This is entirely fair, as the work is intentionally an experimental investigation of whether additional useful information can be derived from models pre-trained with binary auto-labelled data, rather than proposing a new methodology per se.

We would argue that an experimental demonstration of a potential broader use case for the wide range of existing pre-trained models is of high interest to the community and outweighs the noted lack of technical innovation. As noted by reviewer tMQD, we feel that the experimental contribution is of sufficient interest for acceptance as a conference article, and developments on the technical and modelling side would form the basis of further work for conference or journal submission in future.

Point by point responses are attached as comments to each review.

---

### Comment · Area_Chair_9Cjb · 2024-03-18
**Invitation to reply to authors**

Dear reviewers,

The authors have prepared responses to your comments, which you should now be able to see in OpenReview. We encourage you to reply to their comments, and where necessary, adjust your rating. Please do so before the 27th of March.

---

### Meta-Review · Area_Chair_9Cjb · 2024-04-03

**Recommendation:** Accept (Poster)
**Confidence:** 3

**Metareview:**

The paper proposes to train a classifier using image-level binary labels of chest pathology, and use the output logits as a measure of severity of that pathology. This is validated on a prioprietary set with radiologist-assessed severity, and a public dataset but with age labels.

In the initial reviews, the reviewers agree on the relevance of the idea, but as the main issues point out lack of novelty or methodological grounding, the link between the severity and age experiments, and not using public data. After revisions and discussion, two out of three reviewers are not fully satisfied with the responses, and none of the reviewers update their score. On a positive note reviewer one thinks the paper should be presented, and one reviewer comments that the content is not sufficient for a full paper (but in my interpretation, potentially interesting for the conference).

In addition to the discussion, I would like to add some points that might be relevant to consider for the authors:

•	“binary labels can be obtained reliably from the original plain text clinical reports “ – the reliability of this at least in public datasets has been questioned, see for example the work of Lauren Oakden-Rayner and colleagues. In this sense the age experiment is interesting to add since there is less ambiguity in the labels.

•	Training with binary labels but inferring severity from the classifier, with application in chest imaging, has been explored somewhat in papers on multiple instance learning, for example “Detecting emphysema with multiple instance learning” by Ørting et al. There are multiple differences from the current work so I this is not to comment on the novelty, but there could be some connections there worth discussing.

 Although the overall score is below average and there are some shortcomings, writing this meta-review convinces me that this is a good paper to discuss at the conference, as one reviewer also points out.

With this in mind I would recommend for the paper to be accepted if there is enough spots at the conference (this is up to the program chairs), and otherwise I would strongly recommend the authors to submit a short paper to still join the discussion.

---

### Decision · Program_Chairs · 2024-04-05

Accept (Poster)